# Adaptive Laboratory Evolution of *Staphylococcus aureus* Resistance to Vancomycin and Daptomycin: Mutation Patterns and Cross-Resistance

**DOI:** 10.3390/antibiotics12050928

**Published:** 2023-05-18

**Authors:** Vladimir Gostev, Olga Kalinogorskaya, Julia Sopova, Ofelia Sulian, Polina Chulkova, Maria Velizhanina, Irina Tsvetkova, Irina Ageevets, Vladimir Ageevets, Sergey Sidorenko

**Affiliations:** 1Pediatric Research and Clinical Center for Infectious Diseases, Department of Medical Microbiology and Molecular Epidemiology, 197022 Saint Petersburg, Russia; guestvv11@gmail.com (V.G.); kalinogorskaya@bk.ru (O.K.); sulyan1994@mail.ru (O.S.); sps_96@mail.ru (P.C.); i.tsvetik@gmail.com (I.T.); partina-irina@yandex.ru (I.A.); ageevets@list.ru (V.A.); 2Department of Medical Microbiology, North-Western State Medical University named after I.I. Mechnikov, 195067 Saint Petersburg, Russia; 3Center of Transgenesis and Genome Editing, Saint Petersburg State University, 199034 Saint Petersburg, Russia; sopova@hotmail.com (J.S.); velizhanina.me@gmail.com (M.V.); 4Saint Petersburg Branch of Vavilov Institute of General Genetics, Russian Academy of Sciences, 198504 Saint Petersburg, Russia; 5Laboratory of Signal Regulation, All-Russia Research Institute for Agricultural Microbiology, Pushkin, 196608 Saint Petersburg, Russia

**Keywords:** *Staphylococcus aureus*, MRSA, vancomycin, daptomycin, in vitro resistance selection, VISA, hVISA, glycopeptides, lipoglycopepdies

## Abstract

Vancomycin and daptomycin are first-line drugs for the treatment of complicated methicillin-resistant *Staphylococcus aureus* (MRSA) infections, including bacteremia. However, their effectiveness is limited not only by their resistance to each antibiotic but also by their associated resistance to both drugs. It is unknown whether novel lipoglycopeptides can overcome this associated resistance. Resistant derivatives from five *S. aureus* strains were obtained during adaptive laboratory evolution with vancomycin and daptomycin. Both parental and derivative strains were subjected to susceptibility testing, population analysis profiles, measurements of growth rate and autolytic activity, and whole-genome sequencing. Regardless of whether vancomycin or daptomycin was selected, most of the derivatives were characterized by a reduced susceptibility to daptomycin, vancomycin, telavancin, dalbavancin, and oritavancin. Resistance to induced autolysis was observed in all derivatives. Daptomycin resistance was associated with a significant reduction in growth rate. Resistance to vancomycin was mainly associated with mutations in the genes responsible for cell wall biosynthesis, and resistance to daptomycin was associated with mutations in the genes responsible for phospholipid biosynthesis and glycerol metabolism. However, mutations in *walK* and *mprF* were detected in derivatives selected for both antibiotics.

## 1. Introduction

Methicillin-resistant *Staphylococcus aureus* (MRSA) is one of the main nosocomial pathogens, exhibiting a high rate of multidrug resistance. Until the 1990s, vancomycin and teicoplanin were the only therapeutic options for the treatment of serious MRSA infections. In the middle of the 1990s, the first strains with reduced susceptibility to vancomycin (vancomycin-intermediate *S. aureus*—VISA and hetero-VISA) were described, and later, in the 2000s, vancomycin-resistant *S. aureus* (VRSA) was detected in the USA. A meta-analysis by Shariati et al. revealed that the global prevalence of VRSA, VISA, and hVISA before 2010 was 1.2%, 1.2%, and 4%, respectively; after 2010, a 2.0-, 3.6-, and 1.3-fold increase in prevalence was observed, respectively [1].

Nowadays, many options are available for the treatment of infections caused by MRSA with reduced susceptibility to glycopeptides, including lipoglicopeptides, anti-MRSA cephalosporins, oxazolidinones, tigecycline, and daptomycin; however, vancomycin and daptomycin are the only available first-line drugs for the treatment of complicated MRSA infections, including bacteremia. Strains with a high level of resistance to daptomycin are rare; however, the use of daptomycin as an alternative to vancomycin is limited due to the associated resistance of certain strains to both antibiotics [2]. Reports of associated resistance in vitro were followed by reports of daptomycin treatment failure in infections caused by VISA strains and vancomycin failure in cases of daptomycin resistance [3,4,5,6,7].

Daptomycin and vancomycin demonstrate different modes of action and different mechanisms of resistance. The main mechanisms resulting in the decreased susceptibility of VISA/hVISA to vancomycin include thickening of the cell wall and accumulation of free D-Ala-Dla residues, decreased autolysis caused by a change in peptidoglycan biosynthesis, decreased activity of the staphylococcal global regulator decreased lysostaphin susceptibility, and changes in cell wall teichoic acids [8]. Mutations in two-component systems (TCS) regulating cell wall biosynthesis (WalKR, GraSR, and VraSR) and global regulators (Agr and RpoB) are involved in resistance development [9]. Recently, data on resistance were obtained for the *cmk* gene product, which participates in the pyrimidine synthesis pathway [10], the putative formate dehydrogenase Fdh2 [11], and the catabolite control protein E (CcpE), which is the first positive regulator of the tricarboxylic acid cycle [12]. Only a limited number of studies are devoted to the role of genetic background in hVISA/VISA development [13,14].

Daptomycin acts on the cell membrane in a calcium-dependent manner. Daptomycin resistance is conditioned by mutations in the genes that are implicated in membrane phospholipid biosynthesis, including the well-studied gene of multiple peptide resistance factors (MprF) involved in membrane processing. Mutations in MprF are also involved in VISA formation [15] and may be considered a potential common mechanism of daptomycin- and vancomycin-associated resistance. The role of mutations in *cls2* and *pgsA* in daptomycin resistance is also well-known [16]. Uncertainty exists regarding the correlation between changes in glycerol metabolism and the biosynthesis of membrane phospholipids. The involvement of numerous biosynthetic pathways in daptomycin resistance and tolerance development was demonstrated using a proteomic approach [9].

New lipoglycopeptides (telavancin, dalbavancin, and oritavancin) have been approved in the USA and Europe for the treatment of complicated skin and skin-structure infections (SSSIs) [17] and are considered potential antibiotics against vancomycin- and daptomycin-resistant strains. Their advantages over vancomycin against hVISA/VISA isolates are manifested in a strong interaction with the membrane-bound Lipid II precursor of peptidoglycan, membrane insertion of the lipophilic tail (which is similar to the daptomycin mechanism of action), and a weak interaction with decoy targets bearing free D-Ala-D-Ala termini [18,19]. In vitro studies showed that lipoglycopeptides are more active than vancomycin against VISA/hVISA isolates, with minimal inhibitory concentrations of MICs below 2 mg/L; however, the MIC level frequently remains higher than current clinical breakpoints of susceptibility (0.125 mg/L EUCAST breakpoints) [19]. In several case reports, lipoglycopeptide treatment failures were described in VISA-associated infections [20,21,22].

The aim of the study was to determine the impact of the acquisition of resistance to vancomycin or daptomycin on cross-resistance to both antibiotics and susceptibility to lipoglycopeptides and lipopeptides with an assessment of unique and common genetic events.

Here, we found that the in vitro selection of resistance to vancomycin or daptomycin not only leads to cross-resistance between the two mentioned antibiotics but is also associated with resistance to telavancin, dalbavancin, and oritavancin. Resistance to vancomycin is mainly associated with mutations in the genes involved in cell wall biosynthesis, and resistance to daptomycin is associated with the genes involved in phospholipid biosynthesis and glycerol metabolism. Mutations in *walK* and *mprF* were detected in derivatives selected for both antibiotics.

## 2. Results

### 2.1. Phenotypic Changes during Resistance Selection

#### 2.1.1. Changes during Selection with Vancomycin

At the end of selection with vancomycin, the maximum selection concentration reached 32 μg/mL (a detailed protocol for the selection experiment is presented in Figure 1 and the selection steps are presented in Appendix A), and derivative strains demonstrated MICs of 4–8 μg/mL. After 10 passages in antibiotic-free media, the MICs of vancomycin decreased by one dilution. Selection with vancomycin led to an increase in the MICs of the glycopeptide, lipoglycopeptide antibiotics (0.5–4 μg/mL), and daptomycin (2 μg/mL) in derivatives of all strains (Table 1). For certain strains, the seesaw effect was observed. Susceptibility to beta-lactams was recovered in the strain SA0085 after 40 passages in antibiotic-free media due to SCC*mec* excision (change in the oxacillin MIC from 512 μg/mL to 1 μg/mL). Additionally, in this isolate, the MIC level was changed to tetracycline during vancomycin selection. The strain ATCC 29213 recovered its susceptibility to penicillin due to the loss of the plasmid harboring *blaZ* (the MIC data are presented in Appendix A). No changes in the MIC levels of other antibiotics in control strains were observed during passaging in antibiotic-free media.

#### 2.1.2. Changes during Selection with Daptomycin

During selection with daptomycin, the maximum selection concentration reached 128 μg/mL. After the fifth passage, only one strain (SA0422) demonstrated an increased MIC of daptomycin. By the 20th passage, all strains acquired a high level of daptomycin resistance (MIC range: 8–32 μg/mL). Simultaneously, increased MIC levels of vancomycin, teicoplanin, oritavancin, and dalbavancin were also observed by the end of daptomycin selection. The MIC of telavancin was in the range of 0.125–0.25 μg/mL. No changes in the MIC levels of antibiotics were observed in control strains during passaging in antibiotic-free media supplemented with calcium. For the SA0736 strain, by the 40th passage in daptomycin, phenotypic dissociation was observed: white (DAP-W) and yellow (DAP-Y) colonies appeared, and both colony types were analyzed.

### 2.2. PAP Analysis

A population analysis profile (PAP) of vancomycin revealed a significant increase in the AUC of the derivatives compared with the parental isolates. After 40 passages in vancomycin, the AUC ratio was in the range of 1.0–1.2; after daptomycin selection, the AUC ratio was in the range of 0.4–1.0. The dynamic changes in the vancomycin PAP/AUC ratio are presented in Figure 2, and raw plots of PAPs are presented in Appendix A.

### 2.3. Induced Autolytic Activity

All derivative strains were resistant to vancomycin and daptomycin after 40 passages and were characterized by a decrease in their induced autolytic activity in comparison with their progenitors (Figure 3). Prior to in vitro selection, the autolytic activity level was 41.5–69% lysed cells; after vancomycin selection, all derivative strains demonstrated complete resistance to Triton X-100 (0–10% lysed cells, *p* < 0.001). Following daptomycin selection, derivatives showed 0–31% lysed (*p* < 0.01) cells when exposed to Triton X-100.

### 2.4. Growth Kinetics

The growth kinetics of parental strains were compared to those of the derivatives, which were obtained after 40 passages (Appendix A). All strains were characterized by a decreased growth rate. However, the most pronounced growth impairments (decrease in growth rate, increase in doubling time, and lag phase duration) were observed in strains after selection in daptomycin. In control strains, no changes in growth rate were observed. To illustrate the described patterns, the corresponding growth curves are shown in Figure 4 and Appendix A.

### 2.5. Whole-Genome Sequencing Analysis Data

Whole genome sequencing (WGS) was performed for five strains at several checkpoints: before in vitro selection and after 5, 20, and 40 passages (Figure 1). For control strains during passage in antibiotic-free media, WGS was performed after 40 passages. The entire cell biomass that grew on the media with antibiotics was used for sequencing, and the percentage of reads with different mutations was determined. Synonymous mutations and SNPs in non-coding regions were excluded from the analysis. Mixed alleles with an individual SNP frequency of more than 90% of sequence reads were considered homo-mutations (homo-positions). Alleles with an individual SNP frequency in the range of 5% to 90% were considered hetero-mutations (hetero-positions). The correlation of read coverage and the presence of mixed alleles were estimated using all raw sequence data (Appendix A). Hetero-mutations appeared in individual genes at various stages of selection, and as the selection progressed, the population either reverted to the wild type (WT) or the mutation was fixed in the entire population.

During the selection with vancomycin and daptomycin or passaging in antibiotic-free media, various patterns of mutations were observed (Figure 5A–C). Only in derivatives selected with vancomycin and daptomycin were mutations in 49 and 66 loci detected, respectively, and mutations in 20 loci were observed in derivatives after passaging in antibiotic-free media. Mutations in 12 loci were the same in derivatives selected with vancomycin and daptomycin, and three loci mutations were detected in experimental conditions, including passaging in an antibiotic-free medium (overlapped regions in the Venn diagram shown in Figure 5D). All identified mutations and allele frequencies are presented in Appendix A.

### 2.6. Genetic Changes during Selection with Vancomycin

Mutations identified during vancomycin selection were strain-variable. VISA-associated amino acid substitutions (AASs) were identified in different TCSs involved in cell wall biosynthesis (WalK/WalR, YycI/YycH, VraS/VraT, and VraG) and RNA polymerase subunits (RpoB and RpoC) (Figure 6). A mutation in *mprF* was detected in the SA0085 derivative after the 20th passage. In the SA0077 derivative, AASs (G223D and K13R) in WalK were detected after the fifth passage in less than 10% of reads. These AASs were replaced by Y505H at the 20th–40th passage in 100% of reads. Stop codons were detected in the SA0422 derivative in *yycH* after the fifth passage in less than 50% of reads, but at the 20th–40th passages, this mutation was replaced by a 23 bp deletion, which covered 100% of reads. A mutation in *mecA*-foldase (*prsA*) was identified in the SA0085 derivative at the 40th passage (77% of reads). Overall, other identified mutations were detected in different genes involved in different biosynthetic pathways, including phosphate homeostasis (*pitA*), the metabolism of succinate (*sucC* and *pdhC*), the tricarboxylic acid cycle, amino acid biosynthesis, and TCS.

### 2.7. Genetic Changes during Selection with Daptomycin

In all derivatives, mutations associated with daptomycin resistance were detected in the key genes involved in phospholipid biosynthesis (*mprF*, *cls2*, and *pgsA*) (Figure 6). The following different AASs were detected in the MprF: R50H, S136L, S295L, S309L, and L826F. Mutations on *mprF* appeared after the 20th passage and correlated with the increase in MIC of up to 32 mg/L. Only in the SA0422 strain was AAS (S295L) identified after the fifth passage. In all derivatives, AASs in Cls2 (A23V, L52F, D82Y, A280T, T33N, and R295C) were detected at the 20th–40th passages and were associated with a significant increase in MIC. In the SA0077 strain, AAS (D82Y) was detected after the 20th passage; however, this AAS was eliminated at the 40th passage. Mutations in *pgsA* were detected in ATCC29213 and SA0422 derivatives. AAS G170D was not associated with an increase in MIC at the fifth passage. In SA0422, AAS (R579L) in cell wall regulator WalK was detected after the 20th passage.

Several mutations in glycerol metabolism genes were detected. In SA0422, ATCC29213, and SA0736 derivatives, mutations in *fabF* (part of the fatty acid and phospholipid metabolism) and *gerC* (SA0085) were identified. Deletion of a cluster of glycerol metabolism genes (*mutL*, *glpP*, *glpF*, *glpK*, and *glpD*) was identified in ATCC29213.

Deletion in *sepA* was identified in the SA0422 derivative after the 40th passage. During selection with daptomycin, other mutations were detected in different loci of the general metabolism, DNA metabolism, and ion transporters (Appendix A).

Different mutations were associated with a putative protein (SACOL1927 and YfhP), whose role in daptomycin resistance is unknown.

After 40 passages in daptomycin, strain SA0736 dissociated into two lineages, represented by yellow (Y) and white (W) colonies. The Y lineage carried mutations in demethylmenaquinone methyltransferase (*menG*), which is part of the menaquinone biosynthesis pathway and stop gain in *mnhD*, which is involved in Na+ excretion. The W lineage carried mutations in *mtlD*, which is part of mannitol metabolism; in the chromosomal efflux pump (*sepA*), which conferred low-level resistance to disinfectants; and the pyruvate kinase (*pyk*) gene, which is part of the glycolysis pathway. Several genetic changes were identical in both lineages.

### 2.8. Analyzing the Contribution of the Deletion in SACOL1927 (yfhP) to Daptomycin Resistance

A 146 bp deletion and subsequent frameshift in SACOL1927 (*yfhP*) were introduced using CRISPR/Cas9-based genome editing, leaving only seven amino acids from the start codon. The deletion of SACOL1927 (*yfhP*) through allele replacement experiments did not change the MICs of daptomycin, vancomycin, or lipoglycopeptide in the *S. aureus* mutant strain RN4220 (Appendix A).

### 2.9. Genetic Changes during Passaging in Antibiotic-Free Media

Mutations in derivative strains after passaging in antibiotic-free media were represented by mixed alleles in different genes of the central metabolism and other biosynthetic pathways (Figure 5D). We did not find any mutations in biosynthesis pathway membrane phospholipids or in cell wall biosynthesis genes. Other mutations were also identified. Derivative SA0085 in both media (with and without 50 mg/L calcium) recovered beta-lactam susceptibility due to the loss of the SCC*mec* element (*ccr*-complex and *mec*-complex) from the *orfX* region.

## 3. Discussion

In this study, we used the classic selection scheme, including a multistep increase in antibiotic concentration during a relatively long period of 40 passages. All included strains were passaged in parallel, allowing us to evaluate the simultaneous phenotypic and genotypic evolution of *S. aureus* under exposure to two antibiotics. We used long-term in vitro selection over 40 serial passages to achieve a stable and high level of resistance to both antibiotics. Similar approaches were used in previous long-term evolution studies (more than 30 passages) involving exposure to daptomycin [23] or 60 days of vancomycin-based passaging [24]. Long-term selection was accompanied by a high fitness cost [23].

In the current study, well-known mutations in genes related to vancomycin resistance were identified in derivative strains. These include the following genes: WalK, MprF, RpoB, and YycH-YycI. Mutations at the abovementioned loci were also previously observed in daptomycin-resistant strains [15,25,26]. The identified mutations in VraS and VraT were studied in detail and associated with VISA phenotypes [27]. One derivative strain (SA0085) acquired an AAS P314L at the junction between the synthase domain and the flippase domain of MprF during selection with vancomycin. The AAS P314L has previously been shown to cause a moderate increase in daptomycin MIC to 2 g/mL [28], whereas, in another study [29], it had no effect on MIC. In the study of Sulaiman et al. [30], it was demonstrated that mutations in the regulatory gene *yycH* induce tolerance to daptomycin and cause an increase in the MIC of vancomycin. The WalKR is controlled by the YycI-YycH system; therefore, deletions and mutations associated with it have an impact on the development of VISA phenotypes and the decline in autolytic activity [31]. In the current study, all the derivatives were resistant to induced autolysis. Many studies have shown that VISA phenotypes are associated with resistance to induced autolysis, which is one of the biological markers related to VISA/hVISA phenotypes [27]. Staphylococcal autolysins participate in autolysis and promote peptidoglycan homeostasis. Vancomycin may not reach the action site because the cell wall has thickened as a result of autolysin suppression and decreased autolysis in VISA [32,33]. Reduced autolytic activity in daptomycin-resistant isolates suggests that the cell-wall regulon is also involved in daptomycin resistance. To the best of our knowledge, new signatures of vancomycin resistance, which have not yet been described elsewhere, were identified. Potential new markers included WalK (Y505H), RpoB (R406H), YycI (V13L), and RpoC (A949S). The AASs F632S and Q961K in RpoC have previously been associated with in vitro evolved daptomycin resistance [34].

All combinations of mutations obtained in the derivative strains after selection with vancomycin affect resistance to lipoglycopeptides. Several studies have modeled resistance to lipoglycopeptides. In one study [35], mutations in WalK and VraT genes were obtained during dalbavancin selection. Song et al. [36] showed that telavancin selection affects both reduced susceptibility to daptomycin and the formation of the VISA phenotype. Werth et al. [37], after simulating single-dose exposures of dalbavancin to an in vitro PK/PD model for 28 days, described derivative strains with reduced susceptibility to dalbavancin, vancomycin, and daptomycin, primarily due to mutations in the WalKR system. Despite the advantages of these antibiotics, the mechanism of resistance and the pattern of mutations appear to be similar to those observed during the formation of resistance to vancomycin and daptomycin.

Derivative isolates passaging in daptomycin were characterized by the acquisition of well-characterized mutations in MprF, Cls2, and FabF genes. Fatty acid synthase (FabF) is associated with daptomycin resistance and the formation of small colony variants [3]. Mutations in *fabF* were identified in three derivative strains (SA0422, SA0736, and ATCC29213). The *cls* gene is also associated with resistance to daptomycin [38]; however, only one derivative strain in our study had this mutation.

Several new AASs in MprF (S309L), Cls2 (D82Y, A280T, and R295C), WalK (R579L), and the G170D substitution G170D in 3-phosphatidyltransferase (*pgsA*) were identified in daptomycin-resistant derivatives. An entire locus of glycerol metabolism genes was deleted in the ATCC29213 derivative. According to our knowledge, this is the first case of daptomycin resistance described. The metabolism of glycerol involves several enzymatic reactions, the intermediates of which subsequently participate in the formation of membrane phospholipids. According to [28], daptomycin-resistant isolates decreased the expression of genes involved in glycerol metabolism, specifically the *glpF* gene. A decrease or switch to alternative pathways of glycerol metabolism appears to be associated with resistance to daptomycin.

Several derivatives have been identified with mutations in the putative SACOL1927 (*yfhP*) gene. In our previous study [39], mutations in this gene were also identified during daptomycin resistance selection. In genome editing experiments with the laboratory strain *S. aureus* RN4220, we found no effect of deletion in this gene on antibiotic susceptibility. This result implies that SACOL1927 (*yfhP*) does not directly affect daptomycin resistance. The mutation is more of a compensatory one. An amino acid domain from the YdjM superfamily is found in a transmembrane protein that is encoded by the *yfhP*. The LexA protein controls the activity of the proteins in this superfamily, which participate in the SOS response. Other Gram-positive bacteria contain homologues of this protein that are metal-dependent hydrolases with a putative phospholipase function.

Resistance to lipoglycopeptides is affected by the acquisition of daptomycin resistance through mutations in *mprF* + *cls2*. However, we found that all derivative strains (except SA0422) that have no mutations in WalK were susceptible to dalbavancin or telavancin (MIC ≤ 0.125). This suggests that WalK gene mutations play a critical role in the development of lipoglycopeptide resistance.

Furthermore, based on a review of previously published papers, during selection, we identified different mutations in the genes that are responsible for various metabolic pathways and not directly associated with resistance to glycopeptides and daptomycin (Figure 4). These included genes involved in amino acid biosynthesis, protein metabolism, the tricarboxylic acid cycle (TCA cycle), in particular, the pyruvate dehydrogenase complex (*pdhC*), and many other genes with unknown functions. It is worth noting that *pitA*, *pdhC*, and *pykA* were among the top 30 most-mutated genes in vancomycin-adapted strains [40]. A mutation in *pitA*, which encodes a low-affinity inorganic phosphate transporter, was detected in one isolate (SA0422), while only hetero-positions at the 20th passage of selection were found in the other isolate (SA0085). Mutations in this gene were previously identified in a recent study during vancomycin selection [40] and in a study focusing on tolerance to daptomycin [41]. A mutation in *prsA* (*mecA* foldase) was found in SA0085 isolates. A seesaw effect related to daptomycin resistance was previously shown to be accompanied by a decline in β-lactam resistance [42], which may be associated with the impact of *prsA* mutations. As demonstrated in several experimental studies [40,43], metabolic evolution under antibiotic pressure could play a significant role in serving as a compensatory mutational mechanism and/or alternative gain of resistance or tolerance to antibiotics.

Analysis of the entire pool of sequencing reads is frequently undertaken in long-term evolution studies [44]. We used this approach and found that certain mutations were initially formed in a small percentage of the population. In the SA0422 derivative, a stop codon in the *yycH* gene was detected in less than 50% of reads after the 5th passage on vancomycin; by the 40th passage, a 23 bp deletion in this gene was detected in 100% of reads. In the SA0736 derivative, a mutation in the *fabF* gene was detected in 63% of reads after the 20th passage on daptomycin and in 100% of reads after the 40th passage. During selection, SA0736 dissociated into two lineages, with differences in the mutations in menaquinone biosynthesis, Na+ excretion, mannitol metabolism, the chromosomal efflux pump *sepA*, and a mutation located upstream of pyruvate kinase (*pyk*). These examples indicate the presence of a polyclonal population that eliminates or expands during the selection process. Unfortunately, this approach has some limitations since the derivative strains were only sequenced at three data collection points; however, such studies require more of them. Furthermore, when using this approach to analyze genomic data, it is possible to mistake sequence noise for a positive mutation. The fact that many of the detected mutations were not confirmed by mutagenesis experiments is another limitation. It is important to conduct more in-depth research on a number of mutational events, including those affecting general metabolism and various biosynthesis pathways.

## 4. Materials and Methods

### 4.1. Bacterial Strains

Four MRSA strains isolated from patients with different staphylococcal infections in the period 2011–2014 (Appendix A) and the ATCC29213 strain were used for resistance selection. Their vancomycin- and daptomycin-resistant derivatives were included in the study.

### 4.2. Susceptibility Testing

Vancomycin MICs were determined by broth microdilution using cation-adjusted Mueller–Hinton Broth (CA-MHB, Bio-Rad, Marnes-la-Coquette, France) according to ISO 20776-1-2010. For daptomycin testing, CA-MHB was supplemented with Ca^2+^ (with a final concentration of 50 mg/L). The results were interpreted according to recommendations of the European Committee on Antimicrobial Susceptibility Testing (EUCAST v. 12.0). Vancomycin and daptomycin were purchased from Molekula (Darlington, UK), and lypoglycopeptides (oritavancin, dalbavancin, and telavancin) were purchased from Biosynth Carbosynth^®^ (Compton, UK).

### 4.3. Multistep Resistance Selection

Five isolates were passaged in parallel series with increasing concentrations of vancomycin or daptomycin (Figure 6). A serial two-fold broth dilution with vancomycin or daptomycin was prepared in test tubes. Selection with vancomycin was carried out in 2 mL of brain heart infusion broth (BHI; bioMérieux, France) and selection with daptomycin was undertaken in 2 mL BHI-supplemented Ca^2+^ (with a final concentration of 50 mg/L). In the first stage, a series of antibiotic dilutions from half of the MIC to 2xMIC (between 0.5 and 4 g/mL) were prepared. A high inoculum (~10^8^ CFU/mL) was added (0.02 mL) at each antibiotic dilution and incubated at 37 °C until visible growth with complete turbidity of the medium was observed (after approximately 48–72 h) in at least one of the maximum antibiotic dilutions. Subsequently, suspension from the tube with growth in the maximum antibiotic dilution (0.02 mL) was used for transferring cultures onto the fresh media via a series of higher vancomycin or daptomycin concentrations (at least three dilutions were used at each selection step). A total of 40 passages with a stepwise increase in the antibiotic concentration (from 1 to 32 μg/mL for vancomycin and from 1 to 128 μg/mL for daptomycin) were carried out, respectively. The stability of the resistant derivatives was assessed by subculturing on an antibiotic-free medium for 10 passages, followed by susceptibility testing. Additionally, two randomly selected isolates, SA0077 and SA0085, were passaged on an antibiotic-free medium during 40 passages in parallel: BHI and BHI supplemented with 50 mg/L Ca^2+^. MIC measurements, PAP, and whole-genome sequencing were performed at several checkpoints: before selection and after 5, 20, and 40 passages.

### 4.4. Population Analysis Profile

A vancomycin population analysis profile (PAP) was performed according to Pfeltz et al. [45]. Briefly, the prepared inoculum was serially diluted 10-fold in 1.5 mL of saline from 10^8^ CFU/mL to 10^1^ CFU/mL. Serial BHI plates supplemented vancomycin with 11 dilutions: 0-0.5-1-1.5-2-3-4-6-8-12-16 µg/mL were prepared. For each dilution of the antibiotic in the agar, three dilutions of the inoculum were added to the surface of the plate in a volume of 10 µL (spot). Empirically chosen inoculum dilutions were added for a specific antibiotic concentration so that colonies (between 3 and 50 colonies) could be counted in each spot. Three spots of the inoculum were added in a volume of 10 µL for every antibiotic dilution. Inoculated plates were incubated for 48 h at 37 °C. Colonies from each drop were counted by calculating an average number per dilution. The area under the curve (AUC) was calculated in the R base package using the trapezoid function. The strain *S. aureus* Mu50 (ATCC 700699) was used as a reference. The PAP/AUC ratio was calculated as the ratio of the AUC of the tested strain to the AUC of *S. aureus* Mu50. A PAP/AUC ratio ≥ 0.9 was taken for heteroresistance. PAP was performed twice for parental and once for derivative strains after 5, 20, and 40 passages during vancomycin or daptomycin selection.

### 4.5. Measurement of Growth Rate

Growth kinetics were assessed using an Infinite 200 Pro plate reader (Tecan, Grödig, Austria). Optical density at 600 nm (OD600) was measured every 10 min in 96-well plates undergoing orbital shaking. Eight wells per strain were used for measurements. Growth curves were analyzed with the R package [46]; growth rate (r, min^−1^) and doubling time (Dt, min) were calculated, and the lag time was determined as the time period between inoculation and the first increase in the OD600 value. Measurements were performed in triplicate.

### 4.6. Measurement of Induced Autolytic Activity

Cells were grown to an OD600 value of 0.7 and chilled on ice. They were then washed twice with ice-cold phosphate-buffered saline (PBS) at pH = 7.4 and resuspended to obtain an OD600 value of 1.0 in PBS supplemented with 0.05% Triton X-100. The cells were incubated in a 96-well plate at 30 °C while shaking at 142 rpm (with an amplitude of 6 mm), and the rate of autolysis was assessed by measuring OD600 every 10 min for a period of 5 h using a plate reader. Measurements were performed in triplicate.

### 4.7. S. aureus Genome Editing

Deletion in the SACOL1927 (yfhP) gene in the RN4220 strain of *S. aureus* was introduced using the method described by Penewit et al. [47]. Briefly, a spacer sequence for sgRNA was selected in vast proximity to the ATG start codon in the SACOL1927 (yfhP) gene. A 90 bp long primer carrying 5′ phosphorothioate bonds (90bp_sav_del) was used as the ssDNA template for recombination. Two complementary oligos (Gibson_for_SACOL1927 and Gibson_rev_SACOL1927) were used for cloning the protospacer into the sgRNA backbone of the pCas9counter counterselection vector, resulting in the pCas9counter-sav plasmid. Competent cells of the RN4220 strain were transformed with a temperature-sensitive recombineering vector pCN-EF2132tet, encoding recombinase. The obtained transformants were used for a second transformation round with the pCas9counter-sav plasmid and the ssDNA template. Genomic DNA from double transformants was used in the PCR reaction with primers sav_for and sav_rev. The wild-type fragment was 512 bp long, whereas the fragment that underwent deletion was 366 bp long. Transformants with verified deletion were cultured at 37 °C to cure both plasmids. The primer sequence is presented in Appendix A. *Escherichia coli* strains were maintained using LB media, and *S. aureus* strains were maintained using brain-heart agar. *E. coli* carrying ampicillin resistance-encoding shuttle vectors was cultured at 37 °C in the presence of 100 μg/mL antibiotic, *S. aureus* carrying chloramphenicol resistance vectors was cultured at 32 °C in the presence of 10 μg/mL antibiotic, and *S. aureus* carrying erythromycin resistance vectors was cultured at 32 °C in the presence of 10 μg/mL antibiotic. *S. aureus* strains carrying multiple vectors were maintained using both antibiotics. Genome deletion was confirmed by PCR and Sanger sequencing.

### 4.8. Whole-Genome Sequencing

Genomic DNA was extracted using a PureLink™ Genomic DNA Mini Kit (Invitrogen™, Waltham, MA, USA), with preliminary cell lysis performed using 1 mg/mL of lysostaphin (Sigma-Aldrich, St. Louis, MO, USA). Nextera XT or Nextera Flex Kits (Illumina, San Diego, CA, USA) were used for DNA library preparation, followed by sample indexing and amplification according to the manufacturer’s protocol. The concentrations and fragment sizes of DNA libraries were validated with a Qubit Fluorometer using a dsDNA HS Assay Kit (Invitrogen, USA) and a 4150 TapeStation System (Agilent, Santa Clara, CA, USA) with a High Sensitivity DNA ScreenTape Kit, respectively. DNA libraries were sequenced on a MiSeq instrument (Illumina, San Diego, CA, USA) using v3 600-cycle reagent cartridges (Illumina, San Diego, CA, USA).

### 4.9. Bioinformatic Analysis

Reads were filtered and trimmed using Trimmomatic [48]. De novo contigs were assembled with SPAdes [49]. Quality checks of reads and contigs were performed using FastQC [50] and Quast [51], respectively. Reads of the mutant isolates were aligned onto assembled contigs of wild-type (WT) isolates and onto the reference genome COL (CP000046.1) using Bowtie [52] and SAMtools [53]. Allele frequency reads were detected using the mixed allele model available in the software Breseq [54]. The threshold was set to 5%, and all mutational events below this point were excluded. Unassigned new junction evidence was ignored. Sequence reads of strains before selection were aligned against themselves, and the detected single nucleotide polymorphisms (SNPs) were excluded from all data to prevent the identification of false-positive mutations due to the assembly of reference genome errors. All repeat regions, mobile genetic elements, phage-associated loci, and adhesins with homopolymer nucleotide areas were excluded from analysis due to the high rate of nonspecific polymorphisms. Synonymous mutations and SNPs in non-coding regions were also excluded from the analysis.

## 5. Conclusions

Resistance to vancomycin is mainly associated with mutations in the genes involved in cell wall biosynthesis, and resistance to daptomycin is associated with the genes involved in phospholipid biosynthesis and glycerol metabolism. Mutations in WalK and MprF were detected in derivatives selected for both antibiotics, and they are probably the main cause of cross-resistance between vancomycin and daptomycin. Hetero-mutations appeared in individual genes at various stages of selection, and as the selection progressed, the population either reverted to the wild type or the mutation was fixed in the entire population. Regardless of whether vancomycin or daptomycin was selected, most of the derivatives were characterized by a reduced susceptibility to daptomycin, vancomycin, telavancin, dalbavancin, and oritavancin. It is likely that clinical MSSA and MRSA isolates will also show cross-resistance to glycopeptides and lipopeptides.

## Figures and Tables

**Figure 1 antibiotics-12-00928-f001:**
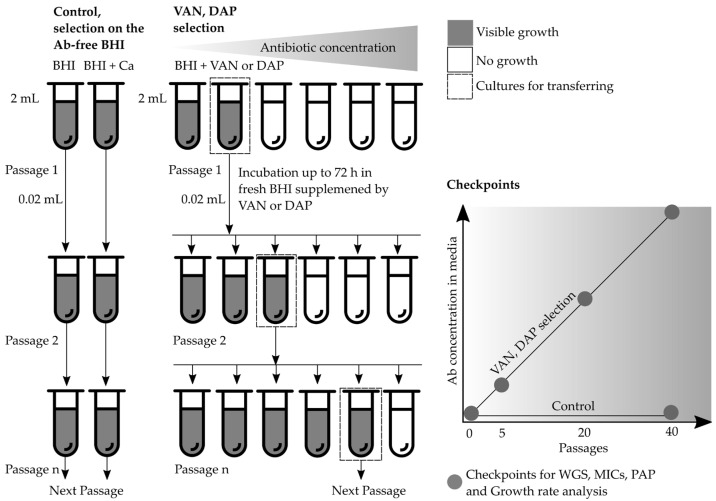
Scheme of adaptive evolution experiments for vancomycin (VAN) and daptomycin (DAP) resistance selection.

**Figure 2 antibiotics-12-00928-f002:**
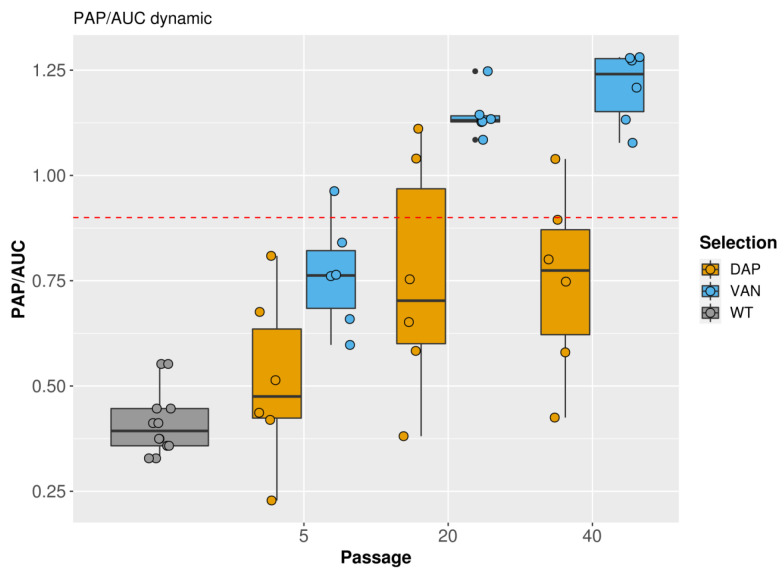
PAP/AUC dynamics of parental isolates (WT) and all isogenic derivatives obtained after 5, 20, and 40 passages during vancomycin (VAN) and daptomycin (DAP) resistance selection. The red dashed line matches the PAP/AUC level and corresponds to the VISA *S. aureus* Mu50 strain.

**Figure 3 antibiotics-12-00928-f003:**
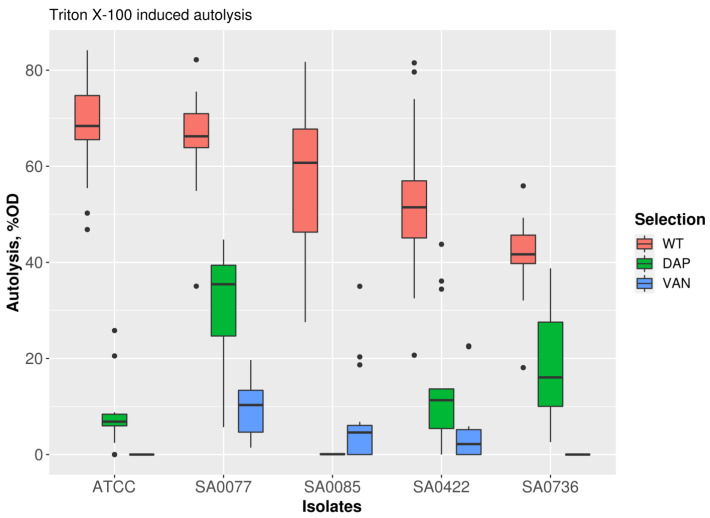
Induced autolytic activity of parental (WT) and isogenic derivatives, obtained after 40 passages during vancomycin (VAN) or daptomycin (DAP) resistance selection.

**Figure 4 antibiotics-12-00928-f004:**
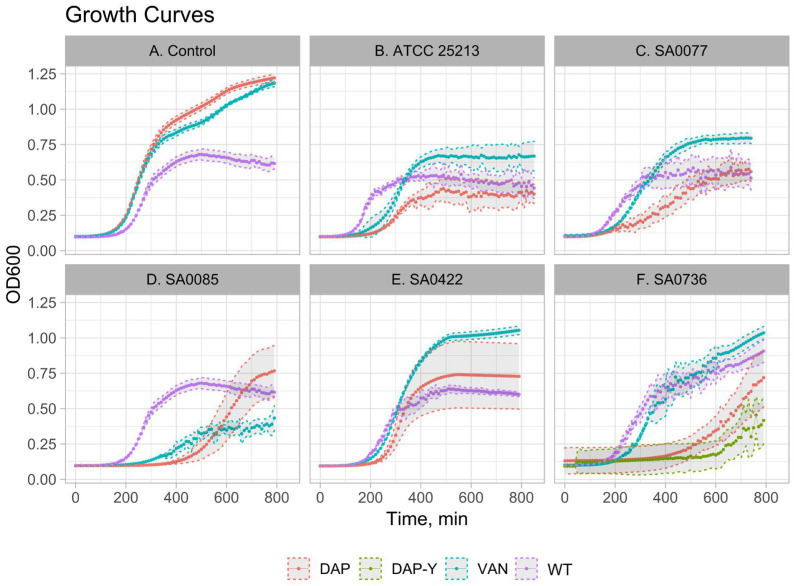
Raw plots of growth curves (mean with standard deviation) are shown. (**A**) Growth curves of control strains, parental strains (WT), and strain after passaging on BHI (VAN) or in BHI + Ca (DAP). (**B**–**F**) Growth curves of parental strains (WT) and derivatives after selection on vancomycin (VAN) or daptomycin (DAP). For strain SA0736 (**F**), growth curves for yellow colonies (DAP-Y) and white colonies (DAP) after selection on daptomycin are shown.

**Figure 5 antibiotics-12-00928-f005:**
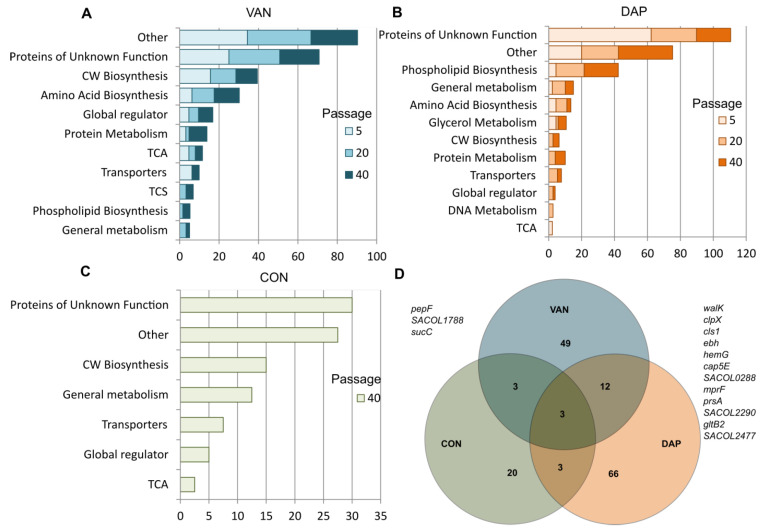
Various pathways for mutations were observed during vancomycin (**A**) and daptomycin (**B**) selection and antibiotic-free passage (**C**) in all strains included in the study from all checkpoints. The cumulative proportion of all mutational events (hetero-mutations and homo-mutations) in the genes of the corresponding systems is shown. Various pathways for mutations which observed during vancomycin (**A**) and daptomycin (**B**) selection and antibiotic-free passaging (**C**). Venn diagram indicating the loci in which mutational events were detected (**D**). Mutations in 12 loci were the same in derivatives selected with vancomycin and daptomycin, and 3 loci mutations were detected in experimental conditions, including passaging in antibiotic-free medium.

**Figure 6 antibiotics-12-00928-f006:**
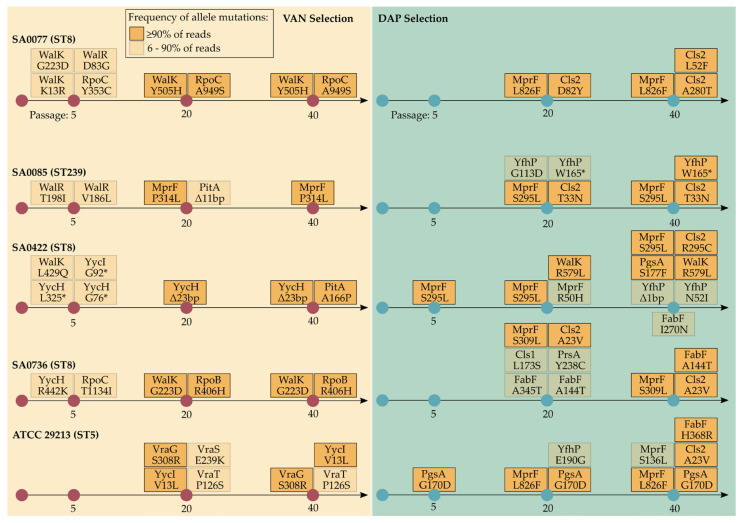
Mutations associated with hVISA/VISA and resistance to daptomycin phenotypes and during selection with vancomycin or daptomycin, respectively.

**Table 1 antibiotics-12-00928-t001:** Susceptibility features of parental strains and their isogenic derivatives during in vitro selection with vancomycin and daptomycin.

Strain	Selecting Agent	Passage	MIC, µg/mL					
			VAN	TEC	DAP	ORI	DAL	TLV
SA0077	VAN	0	1	0.25	0.25	<0.03	0.03	0.125
		5	4	8	2	0.125	0.25	0.06
		20	8	8	2	2	1	0.5
		40	8	8	2	2	2	1
SA0085	VAN	0	1	0.125	0.5	<0.03	<0.016	0.06
		5	4	1	1	0.06	0.06	0.06
		20	4	2	2	2	0.5	0.25
		40	8	8	2	1	0.25	0.25
SA0422	VAN	0	1	0.5	0.25	<0.03	0.03	0.125
		5	2	2	1	0.06	0.125	0.25
		20	4	4	1	0.25	0.5	0.5
		40	4	4	2	0.5	1	1
SA0736	VAN	0	0.5	0.25	0.25	<0.03	0.03	0.125
		5	4	4	0.5	0.25	0.125	0.25
		20	8	8	0.5	2	1	0.5
		40	8	8	2	2	1	0.5
ATCC 29213	VAN	0	0.25	0.06	0.125	0.06	<0.016	<0.03
		5	0.25	0.06	0.25	0.06	<0.016	<0.03
		20	2	2	2	0.06	<0.016	0.125
		40	8	8	2	4	2	0.5
SA0077	DAP	0	1	0.25	0.25	<0.03	0.03	0.125
		5	2	0.125	0.5	2	<0.03	0.03
		20	4	0.25	8	4	0.125	0.125
		40	4	0.25	32	4	0.125	0.125
SA0085	DAP	0	1	0.125	0.5	<0.03	<0.016	0.06
		5	1	0.125	0.5	<0.03	<0.016	0.06
		20	2	8	32	0.06	0.03	0.06
		40	4	8	>64	1	0.125	0.25
SA0422	DAP	0	1	0.5	0.25	<0.03	0.03	0.125
		5	1	0.25	8	0.125	0.06	0.125
		20	2	1	16	0.5	0.25	0.25
		40	2	1	>64	0.5	0.25	0.25
SA0736	DAP	0	0.5	0.25	0.25	<0.03	0.03	0.125
		5	0.5	0.25	1	<0.03	0.03	0.125
		20	4	8	32	2	0.25	0.125
		40W	4	8	>64	4	0.5	0.125
		40Y	4	8	>64	4	0.5	0.125
ATCC 29213	DAP	0	0.25	0.06	0.125	0.06	<0.016	<0.03
		5	1	0.5	0.5	0.06	0.03	<0.03
		20	2	2	32	0.125	0.06	0.125
		40	4	4	64	2	0.125	0.25

## Data Availability

Genomic data have been deposited in the NCBI Sequence Read Archive (SRA), and all reads are available from BioProject PRJNA325350 (SRA run ID: SRR11187837-SRR11187856, SRR5100323-SRR5100339, SRR11809907-SRR11809911, SRR3658373, SRR3658375, and SRR5099566).

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
