# Peer review of "Adaptive Laboratory Evolution of Staphylococcus aureus Resistance to Vancomycin and Daptomycin: Mutation Patterns and Cross-Resistance"

_antibiotics, 2023, doi:10.3390/antibiotics12050928_

Round 1

Reviewer 1 Report

Infections caused by MRSA poses a serious threat to public health, leaving only vancomycin and daptomycin as first-line drugs. The work by Gostev et. al. performed evolution experiments using vancomycin (Van) and daptomycin (DAP) to study how mutations evolved in different strains of S. aureus. They found cross-resistance is common and identified some genetic mutation patterns. Though similar studies have been done before, this work seems to provide some new insights of cross-resistance among different types of antibiotics.  I have some comments below.

1.       Since similar studies have reported and identified some major mutations for Van and DAP, the introduction part may be improved by emphasizing the current gaps and new findings.

2.       This manuscript have many big tables with busy data. It is difficult for readers. Is there a better way to present the data? For example, table 2 could be easily convert to bar graphs, with strains at X-axis and drugs at Y-axis. To me, Fig S3 is much more meaningful than looking at Table  2.

3.       Please clarify and give more detail why and how PAP analysis was done, even though a reference is provided. What was Mu50 strain used as reference?

4.       The authors may need to explain why mutants attain resistance to TX-100 treatment. How does the mechanism connect to resistance to drugs?

5.       Fig1 and Fig 2 captions needs clarification. Specify the mutants strains were from 40 passages.  How many replicates were used?   

6.       Line 159: please specify cultures at varied passages were used for genome sequencing.

7.       Tables 3 and 4: again, these two tables are a bit hard to read at first glance. The dash lines means no mutation detected? Please specify in the note. Also add a note to table 4.

8.       It seems to me that mutations at 40 passages convey the most meaningful information from these tables, if evolution trajectory is not the main point of this study. Mutations are different among strains even under the same selection procedure. It is also interesting that each strain usually only gained a couple of mutations on the selected genes.  I am curious whether these ~2 mutations in a strain are sufficient to cause the resistance observed. The authors might comment on these. Is there a way to experimentally verify the mutation by doing genome editing.  I don’t see any results from experiment in 4.7.

The English looks good to me. 

Author Response

Response to reviewer 1.   

Dear reviewer thank you very much for your thorough analysis of our work and useful comments. Our responses to your comments are below.

Point 1. Since similar studies have reported and identified some major mutations for Van and DAP, the introduction part may be improved by emphasizing the current gaps and new findings.

Response 1. Introduction section was extensively revised.

Point 2. This manuscript have many big tables with busy data. It is difficult for readers. Is there a better way to present the data? For example, table 2 could be easily convert to bar graphs, with strains at X-axis and drugs at Y-axis. To me, Fig S3 is much more meaningful than looking at Table 2.

Response 2. We changed Text Table 2 to Figure S3 (Figure 3 in revised text). Table 2 is move to supplement (Supplemental Table S2 in revised version).

Point 3. Please clarify and give more detail why and how PAP analysis was done, even though a reference is provided. What was Mu50 strain used as reference?

Response 3. PAP is the best method for hVISA/VISA determination, we used this approach to see detect changes during selection on vancomycin or daptomycin. We used only vancomycin in PAP for two reasons. First, we wanted to find out how the level of vancomycin heteroresistance would change during daptomycin selection. And, second, during daptomycin selection, after 5 passages, the daptomycin MIC increased to a high level (more than 2 mg/L, Table 1). In such a case, PAP analysis for low-level heteroresistance detection will be ineffective. As a reference, a well-characterized VISA S. aureus Mu50 strain (ATCC 700699) was used in PAP from our bacterial collection and received from ATCC.

Section 4.4 was revised.

«Vancomycin population analysis profile (PAP) was performed according to Pfeltz et al. [34]. Briefly, the prepared inoculum was serial diluted ten-fold in 1.5 ml of saline from 108 CFU/mL to 101 CFU/mL. Serial BHI plates supplemented vancomycin with 11 dilutions: 0-0.5-1-1.5-2-3-4-6-8-12-16 µ/mL were prepared. For each dilution of the antibiotic in the agar, three dilutions of the inoculum were added on the surface of the plate in a volume of 10 µL (spot). Empirically chosen inoculum dilutions were added for a specific antibiotic concentration so that colonies (between 3 and 50 colonies) could be counted in each spot. Three spots of the inoculum were added in a volume of 10 µL for every antibiotic dilution. Inoculated plates were incubated for 48 hours at 37 °C. Colonies from each drop were count by calculating an average number per dilution. The area under the curve (AUC) was calculated in the R base package using the trapezoid function. The strain S. aureus Mu50 (ATCC 700699) was used as a reference. The PAP/AUC ratio was calculated as the ratio of the AUC of the tested strain to the AUC of S. aureus Mu50. A PAP/AUC ratio >= 0.9 was taken for heteroresistance. PAP was performed twice for parental and once for derivative strains after 5, 20, and 40 passages during vancomycin or daptomycin selection. »        

Point 4. The authors may need to explain why mutants attain resistance to TX-100 treatment. How does the mechanism connect to resistance to drugs?

Response 4. Response 4. Many studies have shown that VISA phenotypes are associated with resistance to induced autolysis, which is one of the biological markers related to VISA/hVISA phenotypes. Amidases and muramidases, two types of staphylococcal autolysins that are involved in autolysis and normally play a role in maintaining the homeostasis of peptidoglycan, control how mature peptidoglycan layers are eliminated throughout the course of a cell's life cycle. Due to autolysin suppression and decreased autolysis in VISA, vancomycin may not reach the action site because the cell wall has thickened. It is noteworthy that we found that DAP-R isolates also showed decreased autolytic activity, indicating that the cell-wall regulon is also involved in the daptomycin resistance. 

Comments were added in the discussion section. Additional references were added.  

«Staphylococcal autolysins participate in autolysis and promote peptidoglycan homeostasis. Vancomycin may not reach the action site because the cell wall has thickened as a result of autolysin suppression and decreased autolysis in VISA [22, 23]. Reduced autolytic activity in daptomycin-resistant isolates suggests that the cell-wall regulon is also involved in daptomycin resistance.»

Point 5. Fig1 and Fig 2 captions needs clarification. Specify the mutants strains were from 40 passages.  How many replicates were used?   

Response 5. Figure 1 and Figure 2 captions were corrected.

Measurements of induced autolytic activity were performed in triplicate (section 4.6) for parental and derivative strains after 40 passages. PAP was performed twice for parental strains and once for derivative strains after 5, 20, and 40 passages.

Section 4.4 was revised.

Point 6. Line 159: please specify cultures at varied passages were used for genome sequencing.

Response 6. The text was corrected. According to the selection protocol, whole genome sequencing was performed at several checkpoints: before selection and after 5, 20, and 40 passages. An additional scheme (Figure 6) was added.   

Point 7. Tables 3 and 4: again, these two tables are a bit hard to read at first glance. The dash lines means no mutation detected? Please specify in the note. Also add a note to table 4.

Response 7. We generate Figure 5 instead of tables 3 and 4 for better reading and understanding of the study results.

Point 8. It seems to me that mutations at 40 passages convey the most meaningful information from these tables, if evolution trajectory is not the main point of this study. Mutations are different among strains even under the same selection procedure. It is also interesting that each strain usually only gained a couple of mutations on the selected genes.  I am curious whether these ~2 mutations in a strain are sufficient to cause the resistance observed. The authors might comment on these. Is there a way to experimentally verify the mutation by doing genome editing.  I don’t see any results from experiment in 4.7.

Response 8. Many mutations were found during the resistance selection process, and many of them (such as genes for general metabolism, transport systems, etc.) were not directly related to vancomycin or daptomycin resistance. Some of them might play compensatory and other supplementary roles in the metabolic support of peptidoglycan or membrane phospholipid biosynthesis that is changing.  The mutations varied between strains, and we hypothesize that this is due to genetic background and prior antibiotic exposure.  We find the question of whether two to three key mutations are sufficient for development resistance in the absence of other discovered mutations to be very interesting. We concur with the reviewer that each mutation should be confirmed using allele replacement technology and an experimental model that is "clean from genetic noise." However, at present, technically, this is very difficult to perform. The study's limitations were added to the discussion section.          

Section 2.8 was added to the manuscript with the results of genome editing. In the discussion section, new comments on results were added.

Reviewer 2 Report

General comments

In the manuscript entitled "In vitro selection of Staphylococcus aureus resistance to vancomycin and daptomycin, mutation patterns and cross-resistance", the authors describe MRSA evolution pathways to get vancomycin and/or daptomycin resistance by using an adaptive laboratory evolution. Although the manuscript contains interesting findings, some clarification is required.

Major specific comments

1.          In the Introduction section, the aim of this study needs to be included. The authors are advised to describe the research question(s).

2.          L. 91 in the Result section, it is difficult to understand the experimental protcol. Hence, the authors are advised to describe an illustration of the experiment protocol as a main figure. It would be informative for readers.

Minor specific comments:

3.          In the Abstract section, what is the difference in terms of terminology between "their resistance (in L. 20)" and "their associated resistance (in L. 20)"?

4.          L. 22 in the Abstract section, the authors are advised to consider rephrasing "in vitro selection" to "adaptive laboratory evolution". Moreover, the title needs to be re-considered.

5.          L. 22 in the Abstract section, it is unclear whether the combined exposure of VAN and DAP was performed in the evolution experiment or two single exposures were applied separately.

6.          In Section 4.3 of the Materials and Methods, it is hard to follow this experiment. In this experiment, were 96 well plates used? Or test tubes? Moreover, how did the authors confirm the bacterial growth? Additionally, how did the author prepare the control group (antibiotic-free group)? Additional information needs to be described.

7.          In Table 1, why the MIC values of ORI are different between SA0077 VAN passage 0 and SA0077 DAP passage 0 (<0.03 and 1)? There are other discrepant cases in the table.

8.          In Tables S1 and S2, each initial MIC value of VAN or DAP for each tested strain needs to be demonstrated.

9.          In Fig. 1, what does the dashed line in red mean?

10.       It is unclear whether the data in Figure 3 expresses WGS analyses for all five strains. An additional explanation is needed in Figure Legend.

Author Response

Response to reviewer 2.

Dear reviewer thank you very much for your thorough analysis of our work and useful comments. Our responses to your comments are below.

Major specific comments

Point 1. In the Introduction section, the aim of this study needs to be included. The authors are advised to describe the research question(s).

Response 1. The following text included: The aim of the study was to determine the impact of the acquisition of resistance to vancomycin or daptomycin on cross-resistance to both antibiotics, susceptibility to lipoglycopeptides and lipopeptides with an assessment of unique and common genetic events.

Point 2. L. 91 in the Result section, it is difficult to understand the experimental protocol. Hence, the authors are advised to describe an illustration of the experiment protocol as a main figure. It would be informative for readers.

Response 2. A suggested reviewer scheme of selection protocol was added in Figure 6 in the Materials and Methods Section. Section 4.3 was revised.

Minor specific comments:

Point 3. In the Abstract section, what is the difference in terms of terminology between "their resistance (in L. 20)" and "their associated resistance (in L. 20)"?

Response 3. Corrected. In the second case we used term “cross-resistance”

Point 4. L. 22 in the Abstract section, the authors are advised to consider rephrasing "in vitro selection" to "adaptive laboratory evolution". Moreover, the title needs to be re-considered.

Response 4. Corrected. New title: Adaptive laboratory evolution of Staphylococcus aureus resistance to vancomycin and daptomycin, mutation patterns and cross-resistance

Point 5. L. 22 in the Abstract section, it is unclear whether the combined exposure of VAN and DAP was performed in the evolution experiment or two single exposures were applied separately.

Response 5. Corrected. To clarify the meaning, we used "or" instead of "and".

Point 6. In Section 4.3 of the Materials and Methods, it is hard to follow this experiment. In this experiment, were 96 well plates used? Or test tubes? Moreover, how did the authors confirm the bacterial growth? Additionally, how did the author prepare the control group (antibiotic-free group)? Additional information needs to be described.

Response 6. Section 4.3 of the Materials and Methods was revised, and an additional scheme was added. Test tubes were used for in vitro selection in a final broth volume of 2 mL.

Bacterial growth was detected with the unaided eye, and complete turbidity of the medium was observed. We did not use additional methods for confirmation of bacterial growth.

Isolates for the control group were randomly selected.

Corrected text below:

«Five isolates were passaged in parallel series with increasing concentrations of vancomycin or daptomycin (Fig. 6). A serial twofold broth dilution with vancomycin or daptomycin was prepared in test tubes. Selection with vancomycin was carried out in 2 mL of brain heart infusion broth (BHI; bioMérieux, France) and selection with daptomycin was undertaken in 2 mL BHI-supplemented Ca2+ (with a final concentration of 50 mg/L). At the first stage, a series of antibiotic dilutions from half of the MIC to 2xMIC (between 0.5 and 4 g/mL) were prepared. A high inoculum (~108 CFU/mL) was added (0.02 mL) at each antibiotic dilution and incubated at 37 °C until visible growth with complete turbidity of the medium was observed (after approximately 48 – 72 h) in at least one of the maximum antibiotic dilutions. Subsequently, suspension from tube with growth in the maximum antibiotic dilution (0.02 mL) was used for transferring cultures onto the fresh media via a series of higher vancomycin or daptomycin concentrations (at least three dilutions were used at each selection step). A total of 40 passages with a stepwise increase in the antibiotic concentration (from 1 to 32 μg/mL for vancomycin and from 1 to 128 μg/mL for daptomycin) were carried out, respectively. The stability of the resistant derivatives was assessed by subculturing on an antibiotic-free medium for 10 passages, followed by susceptibility testing. Additionally, two randomly selected isolates SA0077 and SA0085 were passaged on antibiotic-free medium during 40 passages in parallel: BHI and BHI supplemented with 50 mg/l Ca2+. MIC measurements, PAP, and whole-genome sequencing were performed at several checkpoints: before selection and after 5, 20, and 40 passages.»   

Point 7. In Table 1, why the MIC values of ORI are different between SA0077 VAN passage 0 and SA0077 DAP passage 0 (<0.03 and 1)? There are other discrepant cases in the table.

Response 7. We thank the referee for a careful review of our data. All mistaken MIC values were revised.

Point 8. In Tables S1 and S2, each initial MIC value of VAN or DAP for each tested strain needs to be demonstrated.

Response 8. Initial MIC value of VAN and DAP for each tested strain is indicated in Table S1 in the column WT (MICs of parental strains). In the table S2 (S3 according to current numeration) all detected mutations are mentioned, the table already includes a lot of information, we tried to add the values of the IPC, but this significantly complicated the perception of the data.

Point 9. In Fig. 1, what does the dashed line in red mean?

Response 9. The red dashed line matches the PAP/AUC level and corresponds to the VISA S. aureus Mu50 strain. The caption of Figure 1 was revised.  

Point 10. It is unclear whether the data in Figure 3 expresses WGS analyses for all five strains. An additional explanation is needed in Figure Legend.

Response 10. Yes, in Figure 3 (in the revised version, Figure 4) is presented a cumulative analysis from all WGS data for five isolates from all checkpoints. The figure caption was revised.  

Round 2

Reviewer 2 Report

All the points raised in the first version of the manuscript have been properly addressed by the authors.